# Low-Intensity Blue Light Exposure Reduces Melanopsin Expression in Intrinsically Photosensitive Retinal Ganglion Cells and Damages Mitochondria in Retinal Ganglion Cells in Wistar Rats

**DOI:** 10.3390/cells12071014

**Published:** 2023-03-26

**Authors:** Natalia Ziółkowska, Bogdan Lewczuk, Natalia Szyryńska, Aleksandra Rawicka, Alla Vyniarska

**Affiliations:** 1Department of Histology and Embryology, Faculty of Veterinary Medicine, University of Warmia and Mazury in Olsztyn, Oczapowskiego 2, 10-719 Olsztyn, Poland; lewczukb@uwm.edu.pl (B.L.); natalia.skiepko@uwm.edu.pl (N.S.); 2Policlinico Veterinario Rome Sud, 00173 Rome, Italy; o.wiczkowska@gmail.com; 3Department of Pharmacology and Toxicology, Faculty of Veterinary Medicine, Stepan Gzhytskyi National University of Veterinary and Biotechnologies, 097-063-35-12 Lviv, Ukraine; alla.wynjarska@gmail.com

**Keywords:** melanopsin, intrinsically photosensitive ganglion cells, blue light, retina, retinal damage

## Abstract

This study investigated the effect of low-intensity blue light on the albino Wistar rat retina, including intrinsically photosensitive retinal ganglion cells (ipRGCs). Three groups of nine albino Wistar rats were used. One group was continuously exposed to blue light (150 lx) for 2 d (STE); one was exposed to 12 h of blue light and 12 h of darkness for 10 d (LTE); one was maintained in 12 h of white light (150 lx) and 12 h of darkness for 10 d (control). Melanopsin (Opn4) was immunolabelled on retinal whole-mounts. To count and measure Opn4-positive ipRGC somas and dendrites (including Sholl profiles), Neuron J was used. Retinal cryosections were immunolabeled for glial fibrillary acid protein (GFAP) and with terminal deoxynucleotidyl transferase dUTP nick-end labelling for apoptosis detection. LTE reduced the length of Opn4-positive ipRGC dendrites (*p* = 0.03) and decreased Opn4-immunoreactivity in ipRGC outer stratifying dendrites. LTE and STE decreased the complexity of dendritic arborization (Sholl profile; *p* < 0.001, *p* = 0.03, respectively), increased retinal GFAP immunoreactivity (*p* < 0.001, *p* = 0.002, respectively), and caused outer segment vesiculation and outer nuclear layer apoptosis. Ultrastructural analysis showed that LTE damaged mitochondria in retinal ganglion cells and in the inner plexiform layer. Thus, LTE to low-intensity blue light harms the retinas of albino Wistar rats.

## 1. Introduction

Blue light is a component of visible light, and because of its shorter wavelength, it has the highest energy of the visible light spectrum [1]. Although blue light can regulate many physiological processes and can be used to treat circadian and sleep disfunctions, it can also cause insomnia and photochemical damage to the retina when it is used extensively [2,3,4,5,6,7,8,9]. Thus, researchers, ophthalmologists, and the public have raised concerns about heavy use of devices with light-emitting diodes (LEDs), such as tablets, smart phones, and organic LED screens. 

In experimental studies on rodents, exposure to high-intensity, blue-enriched LED light has caused photoreceptor apoptosis, necrosis, and retinal gliosis [9,10]. Blue-light-induced photoreceptor damage is mediated by the photopigment rhodopsin, which increases photon catch capacity and is involved in photochemical damage of the retina in rats [2]. Exposure to blue-enriched LED light at 500 lx for 24 h induces retinal damage in albino rats but not in pigmented rats [4]. Similarly, exposure of albino rats to blue light at ~250 lx causes photoreceptor apoptosis [9]. However, it is unknown whether exposure to lower-intensity blue light will also cause retinal damage in albino rats. This is an important question because most environmental exposures are to low-intensity blue light.

Another photopigment, melanopsin (Opn4), which is found in intrinsically photosensitive retinal ganglion cells (ipRGCs), is sensitive to blue light [11]. However, it is unclear whether Opn4 mediates light damage to ipRGCs, which comprise a small population of retinal ganglion cells (RGCs), and depending on their type (M1–M6), are responsible for non-image-forming, light-dependent responses such as the pupillary light reflex, circadian photoentrainment, neuroendocrine regulation, regulation of sleep–wake states, photophobia, and negative masking of locomotor activity [12,13,14,15]. There is increasing evidence suggesting that Opn4 in ipRGCs is involved in activating image-forming visual pathways and regulating brightness discrimination and contrast detection [16,17,18]. Thus, ipRGCs act both as photoreceptors that respond directly to light and as typical neurons that integrate synaptic input from other retinal neurons and transmit those signals to the brain [19,20]. 

Exposure of albino and pigmented rats to white light decreases the number of ipRGCs, due to Opn4 downregulation [21,22,23], and exposure of albino rats to white LED light changes Opn4 localization in ipRGCs without causing their death [24]. Continuous exposure of pigmented rats to blue light at 1000 lx for 2 d substantially reduces the total area of Opn4-positive nerve fibers, decreases the number of Opn4-positive ipRGCs, damages the mitochondria of ipRGCs, and causes massive photoreceptor apoptosis. Similarly, exposure of pigmented rats to alternating 12 h periods of blue light (1000 lx) and darkness over 7 d also substantially reduces Opn4 expression in ipRGCs but does not damage photoreceptors [25]. However, it is unknown how exposure to low-intensity blue light affects ipRGCs in albino rats, whose retinas are more susceptible to light damage than those of pigmented rats [4,26]. 

There are number of studies demonstrating that short-wavelength light (400–470 nm) is absorbed by enzyme complexes of the electron transport system in mitochondria and that it negatively affects mitochondria function in photoreceptors and RGCs [27,28,29,30,31]. Moreover, exposure to blue light inhibits mitochondrial enzymes related to the oxidative cycle, which can lead to synthesis of reactive oxygen species that are toxic to the retina [28,31]. Thus, it seems important to investigate whether exposure to low-intensity blue light will affect the mitochondria present in RGCs, which contain a large number of these organelles [32,33]. 

Therefore, our objective was to investigate the effects of short-term (2 d) and long-term (10 d) exposure to low-intensity blue light (150 lx) on the retinas of Wistar rats. Specifically, we investigated the effect of exposure to this kind of light on the expression of Opn4 in ipRGCs and on retinal morphology in whole-mount retinas. Additionally, although electron microscopy does not differentiate between ipRGCs and other RGCs, we used this technique to investigate the ultrastructure of RGCs for better insight into the effect of this exposure on the morphology of the RGC layer.

## 2. Materials and Methods

### 2.1. Animals and Experimental Design

Twelve-week-old Wistar rats of both sexes were used in the experiments. Before the experiment, all rats were maintained in 12 h of white fluorescent light (100 lx) and 12 h of darkness, and to exclude possible eye abnormalities, all rats had an eye exam. After 12 weeks, nine rats were maintained for 24 h in constant darkness and then exposed continuously for 48 h to blue light at 150 lx (short-term exposure group; STE); nine rats were exposed to 12 h of blue light (150 lx) and 12 h of darkness for 10 d (long-term exposure group, LTE); and nine rats were maintained in 12 h of white fluorescent light (150 lx) and 12 h of darkness for 10 d (control group). The rats were randomly assigned to these groups, and all rats completed the study. To eliminate the pupillary light reflex, all rats were administered 1% Tropicamide eyedrops (Polfa Warszawa, Poland) two times a day.

All animals were kept in specially designed cages (Appendix A). Blue light was emitted by strips of LED light (463 ± 10 nm). The strength of illumination with white fluorescent and blue light was measured with a standard light meter (Standard ST-8820 Environmental Meter). Additionally, to determine the power of the blue LED light, a laser power meter was used (Power Meter Gigahertz), which indicated 70 µW/cm^2^. The irradiance value for blue light was 3.8 W/m^2^. The details of the experimental designs and the experimental system are presented in Appendix A. The rats were euthanized immediately after light exposure (at 08:00 am) in a CO_2_ chamber. The rats’ eyes were then used for various assays, as shown in Table 1. All experimental procedures were performed in accordance with Ukrainian animal welfare regulations and were reviewed and approved by the Local Ethical Commission at Poltava National Agricultural University (9 July 2021, N1_9.07.2021).

### 2.2. Immunostaining

Retinal whole-mount preparation and immunostaining for Opn4 were performed according to Ziółkowska et al. [25]. To visualize glial fibrillary protein (GFAP), one retinal cryosection (12 µm) from each rat was washed with 0.1 M phosphate buffer and incubated overnight with anti-GFAP primary antibody (G6171, Merck, Saint Louis, MO, USA). After three rinses in 0.1 M phosphate buffer, the samples were incubated with secondary antibody (1:10 000 dilution, Alexa 488, Thermo Fisher Scientific, Waltham, MA, USA) in a 0.1 M phosphate buffer for 3 h. For staining nuclei, DAPI (Merck, Saint Louis, MO, USA) was used at a concentration of 100 µg/mL. The intensity of GFAP staining relative to DAPI on three retinal cryosections from each rat was measured with ImageJ.

### 2.3. Counting of ipRGC Somas and Determination of Opn4-Positive Neural Process Lengths

For manually counting Opn4-positive ipRGC somas in complete whole-mounts, Image J 1.51n was used. For determination of neural process lengths, four randomly selected images from each temporal retina were apposed (1.28 mm^2^ composite image), as were four randomly selected images from each nasal retina (total area of 2.56 mm^2^). Then, to manually trace Opn4 immunoreactive nerve fibers, Neuron J34 for Image J was used, and the length of all these fibers in the 2.56 mm^2^ area was summed. For manual counting of ipRGC somas in this area, Image J was also used.

### 2.4. Sholl Analysis

From each 2.56 mm^2^ area from each rat, 10 to 25 Opn4-positive ipRGCs (the same ones that were used for determining neural process lengths) were used for Sholl analysis. The number of Opn4-positive ipRGCs that could be analyzed differed between groups because experimental conditions affected this number. In this analysis, concentric rings were drawn around the center of each soma with radii ranging from 10 to 110 µm at intervals of 10 µm. The number of Opn4-positive dendritic intersections at each radius were counted with the Sholl plug-in for ImageJ.

### 2.5. Electron Microscopy

Retinal sample preparation for electron microscopy was done as described previously [25]. Briefly, tissue collection (dorsal half of retinas) took place in the first three minutes after heart stoppage. Tissue fixation (2 h, 4 °C) was performed with 1% paraformaldehyde (Merck, Saint Louis, MO, USA) and 2.5% glutaraldehyde (Polisciences, Warrington, PA, USA) in 0.2 M phosphate buffer (pH 7.4). To wash and postfix samples, 2% osmium tetroxide (Polisciences, Warrington, PA, USA) was used (2 h, room temperature). Samples were embedded in Epon 812. Toluidine blue (Merck, Saint Louis, MO, USA) was used to stain semi-thin sections, which were examined to identify regions for electron microscopy. For contrasting of ultrathin sections, uranyl acetate and lead citrate (Polisciences, Warrington, PA, USA) were used, then the sections were examined with a Tecnai 12 Spirit G2 BioTwin transmission electron microscope (TEM, FEI, USA). For each retina, five to seven cross-sections were examined by TEM (Table 1).

### 2.6. TUNEL Assay

To detect apoptosis, a TUNEL assay was done with an In Situ Cell Death Detection Kit (Roche Diagnostics GmBH, Mannheim, Germany) following the instructions of the manufacturer. From each retina, eight cross-sections were examined (Table 1). In a 2000 µm long area of each section, manual counting of TUNEL positive nuclei was performed.

### 2.7. Statistical Analyses

Before analysing the lengths and counts of Opn4-positive neurons and the intensity of GFAP immunoreactivity, the data were checked to verify whether the assumptions of normality and homogeneity of variance were met. This was done using normal quantile–quantile plots and scale–location plots, respectively. These inspections indicated that the data violated the assumption of homogenous variance; thus, to ensure accurate results, a modern robust method based on 20%-trimmed means (Wilcox’s generalization to trimmed means of Dunnett’s T3 method [34,35]) was used to analyse these differences between groups. Because the mean counts of apoptotic nuclei in the retinas of control and STE rats were less than ten and the counts were underdispersed, quasi-Poisson regression was used to analyze this data followed by pairwise comparisons (Tukey’s method for *p*-value adjustment). For analysis of Sholl data, the number of intersections at each radius was counted. Then, to account for the correlation between neurons from the same rat and the different number of neurons from each rat, a mixed model with a random intercept for each rat followed by Tukey’s method (with Satterthwaite degrees of freedom) were used to compare the area under the curve (calculated via natural cubic spline interpolation) between the three groups. Diagnostic plots showed that the data was approximately normally distributed with homogenous variance, and influence diagnostics indicated that no outliers had an undue influence on the results. For all statistical analyses, *p* < 0.05 was considered significant. Statistical calculations were performed using the WRS (v. 40 [35]), emmeans (v. 1.8.2, [36]), and lme4 (v. 1.1–31, [37]) packages for R (v. 4.2.1 [38]).

## 3. Results

### 3.1. Number of Melanopsin Positive ipRGC Somas in Rat Retinas Was Lower after Blue Light Exposure

Although the differences were not statistically significant, the mean numbers of Opn4-positive ipRGC somas in the two blue light exposure groups were about two-fold lower than the mean number in the control group (Figure 1 and Figure 2A–C). Interestingly, the variation between individuals in the control group was much larger than that in the blue light exposure groups (standard deviations: control, 650; LTE, 139; STE, 222).

### 3.2. Length of Melanopsin Positive ipRGC Dendrites Was Shorter after Blue Light Exposure

The mean length of Opn4-positive ipRGC dendrites was 1.7-fold shorter in the LTE group than in the control group, and this difference was statistically significant (Figure 3). The mean length of these dendrites was 1.8-fold lower in the STE group than in the control group. However, even though this difference was larger than the difference between the LTE group and the control group, it was not statistically significant because the variation between individuals was greater in the STE group than the LTE group. The 95% confidence intervals (CIs) for both comparisons with the control group extended from close-to-zero difference to differences of over 75,000 µm.

### 3.3. Exposure to Blue Light Changed Melanopsin Distribution in ipRGCs

The intensity of staining in the somas and dendrites was less intense in the LTE group than in the control group (Figure 2D–F). The reduction of Opn4-immunoreactivity was visible mostly in the outer stratifying dendrites (Figure 2F). In some neurons in the STE group, the staining was also less intense than in the control group. In all three groups, Opn4-positive varicosities were visible on the dendrites, but in the LTE group, they were less prominent and less frequent (Figure 2D–F). In most dendrites in the LTE group, the staining did not extend as far from the soma as it did in the other groups (Figure 2F).

### 3.4. Blue-Light Exposure Reduced the Complexity of the Dendritic Arbor of ipRGCs

In the control group, the mean number of Opn-4 positive dendritic intersections increased steadily up to a distance of 110 µm from the center of the soma. (Figure 4A). In the LTE group, the mean number of Opn-4 positive intersections increased more slowly and began to level off at a distance of 90 µm from the soma center. The Sholl profile of the STE group was intermediate between those of the control and LTE groups. The area under the curve of the Sholl profiles differed significantly between the three groups (Figure 4B).

### 3.5. Retinal Damage after Exposure to Blue Light

Signs of retinal damage were seen in the LTE and STE groups but not in the control group (Figure 5). In the LTE group, chromatin condensation and apoptotic bodies were seen in the ONL (Figure 5C). In the retinas of the STE group, similar signs of apoptosis were occasionally seen in the ONL. In the STE samples, the inner photoreceptor segments appeared normal but were sparser than in the control-group samples (Figure 5B). In the LTE group, those segments were shorter and smaller than in the control group (Figure 5C). Loss of the photoreceptor outer segments was clearly visible in the LTE and STE groups. In the LTE and STE groups, the outer segments were vesiculated (Figure 5B,C). Finally, the ONL appeared to be thinner in the LTE group than in the control group.

### 3.6. Blue-Light Exposure Increased GFAP Immunoreactivity and Changed Its Distribution in Rat Retinas

When assessing the strength of immunoreactivity, it is important to keep in mind that it should be assessed relative to a baseline intensity of staining, because simple visual inspections can be misleading. Therefore, readers are directed to Figure 6 for the distribution of GFAP immunoreactivity and to Figure 7 for the intensity of its immunoreactivity relative to that of DAPI. In these figures, it can be seen that GFAP immunoreactivity was strongest and most widely distributed in the retinal cross-sections of the LTE group. In this group, GFAP immunoreactivity was 13.5-fold stronger than in the control group and 10.3-fold stronger than in the STE group (Figure 7), and it extended from the NFL through the GCL, IPL, and INL to the ONL (Figure 6C). In the STE group, GFAP immunoreactivity was 1.3-fold stronger than in the control group (Figure 7), and it extended from the NFL to the INL (Figure 6A–C). In the control group, GFAP immunoreactivity was present almost exclusively in the NFL and GCL.

### 3.7. Increased Apoptosis in ONL of Rats Subjected to LTE and STE to Blue Light

In control retinas, only isolated apoptosis signals were visible (mean = 1.3 nuclei) (Figure 8A, Appendix A). STE to blue light significantly increased the number of positively labelled nuclei in the ONL 5.9-fold (Figure 8B and Figure 9), and LTE significantly increased this number 33.1-fold (Figure 8C and Figure 9).

### 3.8. Electron Microscopy Revealed Mitochondrial Swelling after LTE to Blue Light

In the STE and control groups, although empty spaces were occasionally present between cells, the ONL appeared normal (Figure 10A,D). However, in the LTE group, signs of apoptosis in the ONL (pyknosis and condensation of the cytoplasm) were occasionally seen (Figure 10G). In this group, cells displaying signs of apoptosis were scattered throughout the ONL and were observed in the outer and inner regions of this layer. In the LTE group, thick processes with intermediate filaments (likely from Müller cells) were observed around cells in the ONL (Appendix A), but these processes were not observed in the STE and control groups.

In the LTE group, the photoreceptor inner segments were sparser than those in the control group (Figure 10B,H); many of these segments were also thinner and longer; and some of the segments were swollen and had brighter cytoplasm (Figure 10H). In the swollen segments, the mitochondria had normal cristae, but the intermembrane space was dilated. In the STE group, the inner segments were sparser but most of them looked normal (Figure 10E).

Photoreceptor outer segments were disarranged and much less numerous in the LTE and STE groups than in the control group. In the LTE group, many of the photoreceptor outer segments had lost their lamellar structure and were vacuolized (Figure 10I). In STE and LTE groups, the apical parts of those segments formed round or ellipsoidal structures containing tubules and vesicles (Figure 10C,F,I). In the LTE group, the outer segments were usually detached from the inner segments (Figure 10H).

In the LTE and STE groups, but not in the control group, mitochondria in retinal RGCs were swollen and had disrupted cristae (Figure 11A–C). These mitochondrial abnormalities were also often seen in the IPL of the LTE group and occasionally seen in this layer in the STE group (Figure 11D–F). In the LTE group, but not in the STE and control groups, many of the nerve processes in the INL were swollen (Figure 11G–I).

Müller cell processes with intermediate filaments were present between the photoreceptor outer segments in all three groups (Figure 10B,E,H). In the LTE group, but not in the STE and control groups, thick Müller cell processes were present between cells in the ONL (Appendix A).

## 4. Discussion

Our results indicate that long-term exposure (LTE) of Wistar rats to blue light at low-intensity (150 lx) reduces the total length of their melanopsin (Opn4)-positive ipRGC dendrites, and that both LTE and short-term exposure (STE) reduce the complexity of Opn4-positive dendritic arborization in ipRGCs. These results also suggest that LTE and STE may reduce the number of Opn-4 positive ipRGC somas. The results indicate that LTE damages photoreceptors in retinas and that LTE and STE increase the number of apoptotic cells in the outer nuclear layer (ONL). Moreover, these kinds of blue light exposures increase GFAP immunoreactivity in the retina and change its distribution. Finally, our results also indicate that LTE to this kind of light does not cause apoptosis in the ganglion cell layer (GCL) but causes mitochondrial swelling in retinal ganglion cells (RGCs) and in the inner plexiform layer (IPL).

We found that the mean number of ipRGCs in the retinas of the control rats was approximately 1600 cells per whole-mount, which is similar to counts in other studies [39,40]. Although the results were not statistically conclusive, our data suggest that, immediately after STE and LTE to blue light at 150 lx, the number of ipRGCs somas in whole-mount retinas is substantially reduced. If the number is indeed reduced, it could recover to normal levels because Garcia-Ayuso et al. [39] reported that one month after exposure to cool white LED light at 3000 lx, the numbers of ipRGCs were similar in exposed and control retinas.

Our results show that LTE to blue light at low intensity reduces the total length of the Opn4-positive areas of dendrites and reduces the density of Opn4-positive varicosities on the dendrites. Moreover, both LTE and STE reduce the complexity of Opn4-positive dendritic arborization in ipRGCs, as shown by substantial reductions in the area under the curve of the Sholl profiles. Our results also suggest that STE (2 d) shortens the length of Opn4-positive areas of dendrites, but the results are not statistically conclusive. In our previous study [25], we found that exposure to high-intensity (1000 lx) blue light for 2 and 7 d causes more substantial reductions in the total length of the Opn4-positive areas of dendrites in pigmented rats than the exposure in the present study. This difference between studies is likely due to the rat strains that were used (non-pigmented vs. pigmented), the intensity of the light, and the amount of damage that the retinas sustained.

The reduction in the length and complexity of Opn4-positive areas of dendrites suggests that light-induced damage to photoreceptors may influence the expression of Opn4 in ipRGCs, as it has been demonstrated that those cells are connected to bipolar and amacrine cells, and that rod and cone signals may be capable of modifying the intrinsic light response in ipRGCs [39,40,41,42]. However, Garcia-Ayuso et al. [39] reported that, when albino rats are exposed to damaging white light, the number of immunodetected ipRGCs and the expression levels of Opn4 begin to gradually recover, while photoreceptors continue to die progressively. More studies are needed to clarify the extent to which rods and cones are involved in Opn4 expression in ipRGCs.

One possible explanation for the reductions in the length and complexity of Opn4-positive dendrites and the density of the Opn4-positive varicosities could be downregulation of Opn4 expression. Opn4 expression in rats is regulated by light and darkness, and light has a strong suppressive effect on the level of Opn4 protein expression in the retinas of rats, hamsters, and mice [22,43,44,45]. For example, Opn4 concentration was significantly decreased in rat retinas after 21 d in constant light [23]. This downregulation is probably a protective mechanism because Opn4 produces reactive oxygen species when it is activated by blue light [30,31].

Another possible explanation for the reduced length and complexity of Opn4-positive dendrites and the reduced density of Opn4-positive varicosities could be a change in the distribution of Opn4 immunoreactivity, likely reflecting a change in the distribution of Opn4 itself. Indeed, we found that Opn4 immunoreactivity decreased in the outer stratifying dendrites of the rats in the LTE group. Normally, Opn4 levels are lowest at the end of photophase and highest at the end of scotophase [22], which suggests that Opn4 protein regenerates during the night-time. However, even though our LTE rats were exposed to alternating periods of blue light and darkness, 12 h of darkness was insufficient time for complete Opn4 regeneration, as shown by the decrease in Opn4 expression in this group. We speculate that this is caused by blue light suppressing Opn4 expression to a greater extent than white light, as Opn4 is particularly sensitive to blue light, which is highly energetic [10,46]. Future studies could employ immune-electron microscopy to investigate changes in the distribution of Opn4 at the subcellular level.

Our findings of decreased Opn4 immunoreactivity in the outer stratifying dendrites of the Wistar rats after blue light exposure and the flattening of the Sholl profile from 90 to 110 µm in the LTE group are consistent with the report of Hannibal et al. [22], who found that Opn4 immunoreactivity progressively decreases in the retinas of albino rats exposed to constant white light. This decrease starts in the distal dendrites and proceeds to the proximal dendrites, until finally, the immunoreactivity is limited to ipRGC somas. Hannibal et al. [23] found that Opn4 expression is reduced primarily in the outer stratifying dendrites of the IPL in pigmented rats exposed to constant white light, without obvious differences in ipRGCs and their proximal dendrites. Similarly, Benedetto et al. [24] showed that Opn4 immunoreactivity decreases in ipRGCs as the time of white-light exposure increases, and that, after 8 d of this exposure, Opn4 immunoreactivity is present in the ipRGC somas but not within the ipRGC dendrites. Hannibal et al. [23] have speculated that shifts in the distribution of retinal Opn4 immunoreactivity after white-light exposure could be a way in which the retina adapts to environmental lighting conditions, maintaining light sensitivity for signal transmission to the brain. The changes that we observed after blue light exposure could have the same function.

A third possible explanation for the reductions in the length and complexity of Opn4-positive dendrites and the density of Opn4-positive varicosities is retinal damage, which was indicated by our finding that LTE to blue light causes apoptosis in the ONL, and that STE also does so, but to a lesser extent.

Our finding that LTE and STE to blue light cause apoptosis in the ONL is an indication that low-intensity blue light exposure damages Wistar rat retinas. This finding is consistent with that of our previous study [25], in which we found massive apoptosis in the ONL of pigmented rats that were continuously exposed to high-intensity blue light for 2 d, although we did not observe damage to photoreceptors in rats intermittently exposed (12 h light, 12 h darkness) to high-intensity blue light for 7 d. The Wistar rats that underwent LTE likely sustained a greater amount of retinal damage than the Brown Norway rats because they have less pigment in the eye. The photoreceptors (both rods and cones) of albino rats are more susceptible to light damage than those of pigmented rats [4,26]. For example, Krigel et al. [4] demonstrated that, after exposure to cold white LED light at 500 lx, ONL thickness was reduced to a lesser extent in pigmented rats than in albino ones. Similarly, Shang et al. [8] demonstrated that rats exposed to blue LED or white LED light at 750 lx for 9 days had significantly reduced ONL thickness. Recent studies have demonstrated that exposure to blue light at 200 lx for 20 s causes progressive (from 1–30 days after exposure) reduction in the outer retina thickness and a reduction in the number of S-cones in albino rat retinas [47]. Although we used even lower intensity blue light (150 lx) in the present study, in our previous study we observed that exposure to blue light at this intensity reduces the thickness of the ONL and the photoreceptor segments [48]. Based on these results with albino rats, we cannot extrapolate with any certainty to humans, but these results suggest that exposure to even low-intensity blue light could be harmful to the retina.

The second indication that low-intensity blue light exposure damages the retinas in Wistar rats is our finding that photoreceptor damage is more extensive after LTE to blue light than after STE. Our results indicate that changes first occur in the apical parts of the outer segments, as seen in both the STE and LTE groups. As exposure to low-intensity blue light continues, the inner segments are affected, and the neural processes in the IPL and their mitochondria swell, as seen in the LTE group. Grimm et al. [2] reported that photoreceptors are damaged by high-intensity blue light and showed that rhodopsin mediates this damage in the rodent retina. Similarly, vacuolization of photoreceptor outer segments was reported in albino rats exposed to low-intensity blue light (150 lx) [48] and in rats exposed to white light [49]. The present findings add to these previous reports by suggesting that the duration of exposure to low-intensity blue light is the most important factor influencing the extent of the damage.

The third indication that retinal damage is caused by low intensity blue light exposure is our finding that LTE substantially increases expression of GFAP in rat retinas, and STE also increases this expression, but to a lesser extent. As GFAP overexpression is proportional to photoreceptor damage after rats are exposed to light, these results indicate that LTE causes more damage than STE [50]. In both blue light exposure groups, GFAP was expressed throughout most or all of the retina. In most vertebrate retinas, in contrast, GFAP is present only in the end foot processes of Müller cells and astrocytes [51]. The increased expression of GFAP in response to blue light exposure indicates hypertrophy of Müller cells, which was evident in electron micrographs, and strongly suggests retinal gliosis.

This finding of increased expression of GFAP after blue light exposure is consistent with the findings of other studies, where GFAP overexpression was caused by different types of retinal injury, including injuries caused by light [3,50,52,53]. Although it had previously been shown that exposure of pigmented rats to high-intensity blue light (8000 lx) [3] caused Müller cell hypertrophy and GFAP overexpression, our study shows that exposure to much less intense blue light causes similar effects. Although retinal neurons are more sensitive to blue-light-induced damage in vitro than Müller cells [54], recent studies have shown that exposure of Müller cells (in vitro) to low-intensity blue light can also cause apoptosis [55].

Our results suggest that both LTE and STE substantially reduce the number of Opn-4 positive ipRGC somas. However, due to the large amount of variation between the individuals in the control group, the 95% confidence intervals for the differences were very wide, extending from close to no difference to a difference of about four-fold lower in the LTE and STE groups. If the number of Opn4-positive ipRGC somas is indeed reduced by exposure to low-intensity blue light, this reduction is unlikely to be caused by cell death, as we did not detect obvious signs of apoptosis in the GCL. However, Huang et al. [56] have shown that long-term exposure of the RGC-5 cell line to blue light (1.3 lx) induces cell death due to apoptosis [56] or necroptosis [57], so signs of apoptosis may have become evident had the exposure in our study been prolonged further.

Interestingly, in situations where light-induced damage is not an issue, ipRGCs are less vulnerable than conventional RGCs, e.g., in cases of optic nerve damage [58] and experimental glaucoma [59]. Additionally, in Leber’s congenital amaurosis, an inherited optic neuropathy, ipRGCs are relatively well preserved. This could be related to the higher content of cytochrome c oxidase and the great abundance of mitochondria in ipRGCs [60]. The resistance of ipRGCs to injury may also be due to the presence of phosphatidylinositol-3 kinase (PI3K) and pituitary adenylate cyclase-activating polypeptide (PACAP) [59,61]. For example, intraocular injection of rats with PI3K inhibitor improved ipRGC survival after optic nerve injury or ocular hypertension [59]. It would be interesting for future studies to investigate if expression of these factors will change during exposure to blue light.

In our study, however, the only sign of ultrastructural damage to RGCs was mitochondrial swelling in the IPL, which was clearly visible in the LTE group and present to a lesser extent in the RGCs in the STE group. This indicates that LTE to blue light damages RGCs in Wistar rats. Similarly, Nunez-Alvarez et al. [57] have demonstrated that blue light negatively affects the function of RGC mitochondria in rats and in the R28 cell line. The mitochondrial swelling that we observed in the present study could have been caused by oxidative stress. Blue light inhibits a key mitochondrial enzyme, which can lead to generation of oxygen species [30,31,57]. A similar explanation for the swelling is the fact that internal mitochondrial membranes contain two chromophores, flavin and porphyrin, both of which have an absorption peak in the blue light range [62]. Light-activated flavins can initiate H_2_O_2_ production, which causes lipid peroxidation [63]. Regardless of its cause, the stress to RGC mitochondria caused by blue light exposure could lead to energy deficits and inhibition of neuronal metabolism and signal transmission. Note that, although standard electron microscopy does not distinguish ipRGCs from other RGs, most RGCs in sections from these groups displayed mitochondrial swelling. To differentiate between RGCs and ipRGCs, future studies could use immunoelectron microscopy.

In assessing our study, three limitations should be kept in mind. The first is the aforementioned inability to distinguish between ipRGCs and other RGCs, which future studies can address as explained above. The second limitation is the small sample size: as suggested by the confidence intervals, is likely that the differences in the number of Opn-4 positive ipRGC somas and the length of Opn-4 positive dendrites between exposure groups would have been statistically significant if our sample size had been larger. Future confirmatory studies can address this issue by using larger sample sizes and/or combining their results with ours via statistical meta-analysis. Finally, it would be interesting for future studies to use quantitative western blotting to investigate how the reduction in Opn4-immunoreactivity in ipRGCs corresponds to likely changes in protein expression.

## 5. Conclusions

In summary, this study shows that exposure to low-intensity (150 lx) blue light reduces the expression of Opn4 in ipRGCs, changes the distribution of Opn4 in these cells, reduces the complexity of their Opn4-positive dendritic arborization, and increases the number of apoptotic cells in the ONL. Exposure to this kind of light does not cause RGC death but damages mitochondria in those cells, which may suggest that RGCs are oxidatively stressed after low-intensity blue light exposure. The mitochondrial damage that was present in the neural processes of the inner retina suggests that signal transmission between the ipRGCs and other neurons of the retina could have been disrupted. Finally, this study shows that exposure to blue light at this intensity for 10 d harms the retina and causes photoreceptor damage. It will be interesting to investigate how the changes in ipRGCs after exposure to blue light that were observed in our study correspond to alterations in their function and the physiological processes that they are involved in (e.g., pupillary light reflex). Thus, our future studies will perform investigations with chromatic pupillometry, electroretinography, and optical coherence tomography under the same conditions.

## Figures and Tables

**Figure 1 cells-12-01014-f001:**
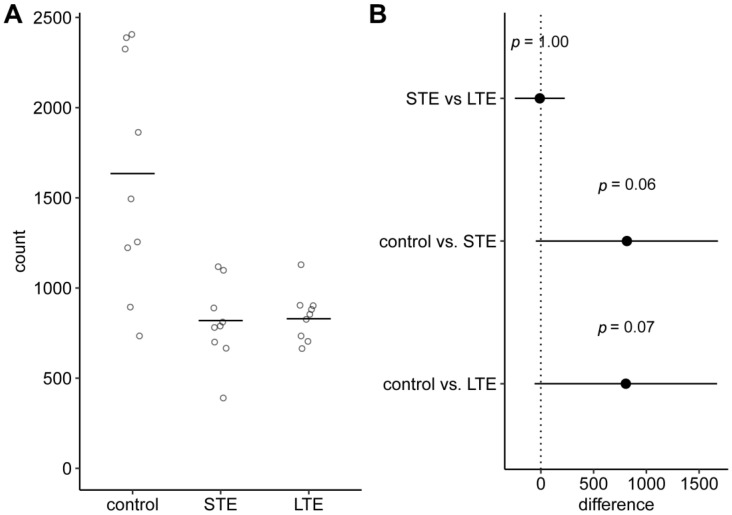
Effect of blue light exposure on number of melanopsin positive intrinsically photosensitive retinal ganglion cells (ipRGCs) in whole-mount retinas from Wistar rats. The rats were exposed to 12 h white light (150 lx) and 12 h darkness for 10 d (control, CTRL; 9 rats), cycles of 12 h blue light (150 lx) and 12 h darkness for 10 d (long-term exposure, LTE; 9 rats), and constant blue light (150 lx) for 2 d (short-term exposure, STE; 9 rats). (**A**) Circles show number of melanopsin positive ipRGCs in a whole-mount retina from one rat. Horizontal lines show 20%-trimmed mean. (**B**) Dot shows difference between group means. Error bar shows 95% confidence interval for the difference. *p*-values and confidence intervals were calculated with Wilcox’s robust technique based on trimmed means [34,35].

**Figure 2 cells-12-01014-f002:**
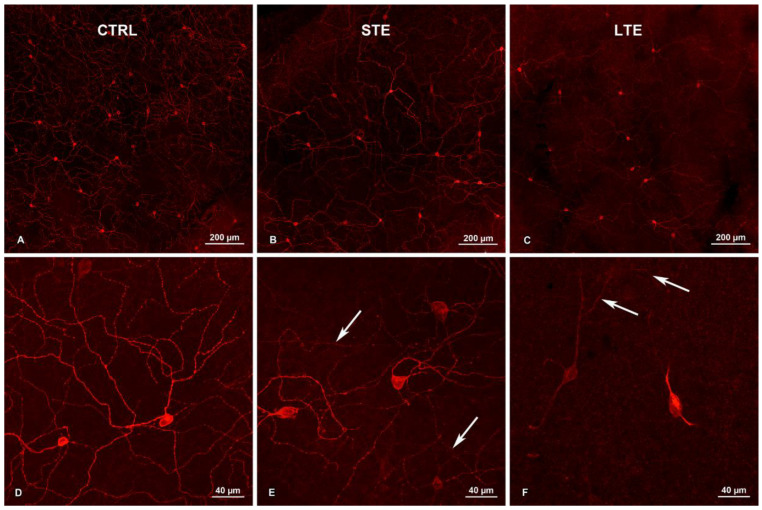
Effect of blue light exposure on melanopsin distribution in intrinsically photosensitive retinal ganglion cells (ipRGCs) in whole-mount retinas from Wistar rats. The rats were exposed to 12 h white light (150 lx) and 12 h darkness for 10 d (control, CTRL; 9 rats), cycles of 12 h blue light (150 lx) and 12 h darkness for 10 d (long-term exposure, LTE; 9 rats), and constant blue light (150 lx) for 2 d (short-term exposure, STE; 9 rats). (**A**) CTRL (20×). Note: dense network of melanopsin-positive dendrites. (**B**) CTRL (40×). Multiple melanopsin-positive dendrites with varicosities are visible. (**C**) STE (20×). A widespread dendritic network of melanopsin-positive processes is visible. (**D**) STE (40×). Multiple melanopsin-positive varicosities are also visible here. Note: immunoreactivity to melanopsin is decreased in outer stratifying dendrites (arrows). Varicosities are also less prominent here. (**E**) LTE (20×). Note: levels of melanopsin expression differ widely in the ipRGC somas. Relative to the other groups, melanopsin immunoreactivity is decreased in the dendrites, and the melanopsin-positive varicosities on the dendrites are less prominent. (**F**) LTE (40×). Melanopsin is present mostly in ipRGC somas and proximal dendrites. Note: lack of varicosities on these processes and decreased melanopsin immunoreactivity in outer stratifying dendrites (arrows). Samples were visualized and photographed with a confocal microscope (LSM900, Zeiss).

**Figure 3 cells-12-01014-f003:**
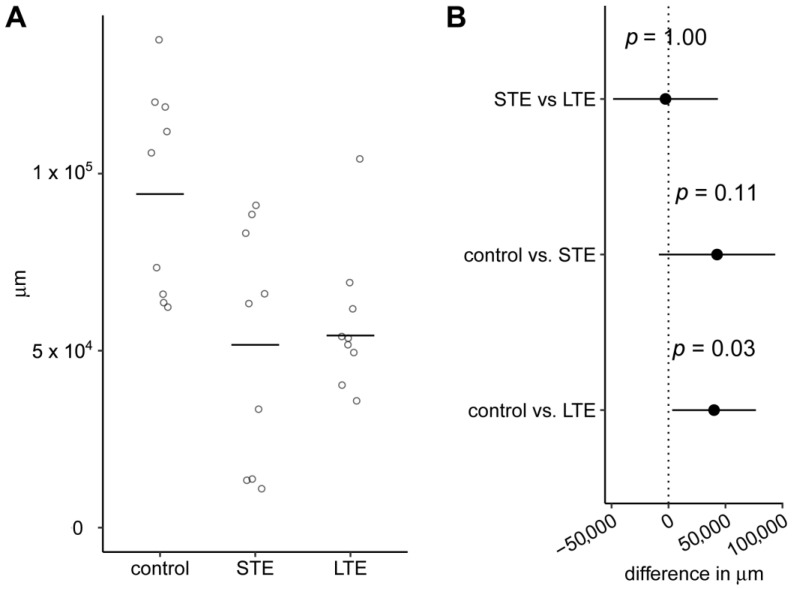
Effect of blue light exposure on length of melanopsin positive intrinsically photosensitive retinal ganglion cell (ipRGC) dendrites. The rats were exposed to 12 h white light (150 lx) and 12 h darkness for 10 d (control, 9 rats), cycles of 12 h blue light (150 lx) and 12 h darkness for 10 d (long-term exposure, LTE; 9 rats), and constant blue light (150 lx) for 2 d (short-term exposure, STE; 9 rats). (**A**) Circles show total length of all melanopsin-positive ipRGC processes in a 2.560 mm^2^ area of a retinal whole-mount from one rat. Horizontal lines show 20%-trimmed means. (**B**) Dot shows difference between group means. Error bar shows 95% confidence interval for the difference. *p*-values and confidence intervals were calculated with Wilcox’s robust technique based on trimmed means [34,35].

**Figure 4 cells-12-01014-f004:**
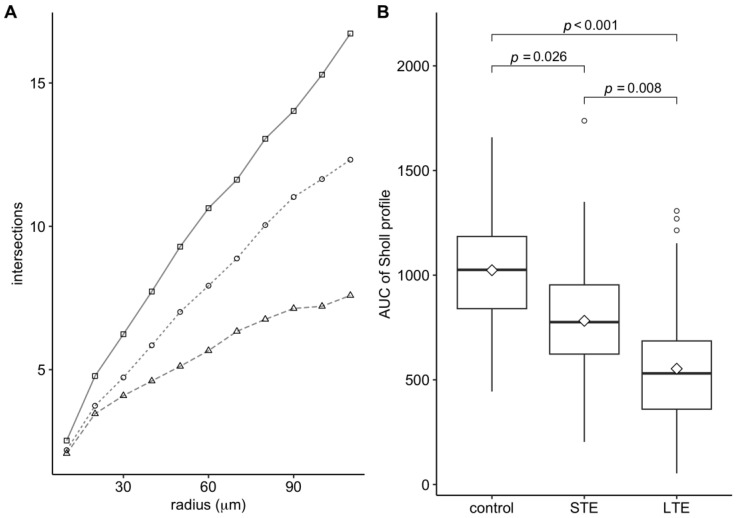
Effect of blue light exposure on dendritic arborization of intrinsically photosensitive retinal ganglion cells (ipRGCs). The rats were exposed to 12 h white light (150 lx) and 12 h darkness for 10 d (control, 9 rats), cycles of 12 h blue light (150 lx) and 12 h darkness for 10 d (long-term exposure, LTE; 9 rats), and constant blue light (150 lx) for 2 d (short-term exposure, STE; 9 rats). (**A**) Sholl profiles: squares, circles, and triangles (control, STE, and LTE groups, respectively) show mean number of melanopsin-positive dendrites intersecting each radius around the ipRGC somas. (**B**) Area under the curve (AUC) of each Sholl profile shown in Figure 4A. The boxes show the central 50% of the data, diamonds show the mean AUC, thick horizontal lines show the median, vertical lines outside the box extend to the furthest non-outlying data points, and circles show outliers. *p*-values were calculated using a mixed model followed by Tukey’s method for pairwise comparisons.

**Figure 5 cells-12-01014-f005:**
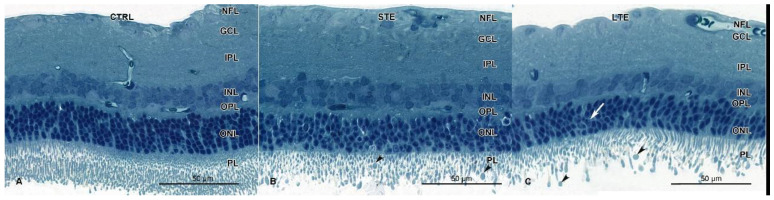
Effect of blue light exposure on the retinal morphology of Wistar rats. The rats were exposed to 12 h white light (150 lx) and 12 h darkness for 10 d (control, CTRL; 9 rats), cycles of 12 h blue light (150 lx) and 12 h darkness for 10 d (long-term exposure, LTE; 9 rats), and constant blue light (150 lx) for 2 d (short-term exposure, STE; 9 rats). Resin-embedded semi-thin sections were stained with toluidine blue. Retinas from the control group (**A**) display normal morphology. Retinas from the STE (**B**) and LTE (**C**) groups show loss of photoreceptor outer segments and their vesiculation (arrowheads). Note signs of apoptosis (chromatin pyknosis and apoptotic bodies) are present in some nuclei of the outer nuclear layer of the LTE group (white arrow). Samples were visualized and photographed with a light microscope (Axioimager, Zeiss).

**Figure 6 cells-12-01014-f006:**
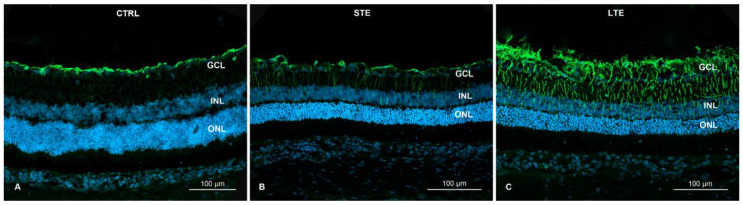
Effect of blue light on glial fibrillary acid (GFAP) protein expression in Wistar rat retinas. The rats were exposed to 12 h white light (150 lx) and 12 h darkness for 10 d (control, CTRL; 9 rats), cycles of 12 h blue light (150 lx) and 12 h darkness for 10 d (long-term exposure, LTE; 9 rats), and constant blue light (150 lx) for 2 d (short-term exposure, STE; 9 rats). (**A**) LTE: Immunolabeling for GFAP in thick Müller cell processes indicating advanced hypertrophy; the immunolabeling extends from the innermost layers of the retina to the outer limiting membrane. (**B**) STE: Immunolabelling for GFAP in Müller cell processes indicating mild hypertrophy. Note thin GFAP-positive fibers extending from innermost layers to the outer plexiform layer. (**C**) GFAP expression in the control retina is limited to the innermost layers (NFL and GCL). Samples were visualized and photographed with a confocal microscope (LSM900, Zeiss). Abbreviations: ONL, outer nuclear layer; INL, inner nuclear layer; GCL, ganglion cell layer.

**Figure 7 cells-12-01014-f007:**
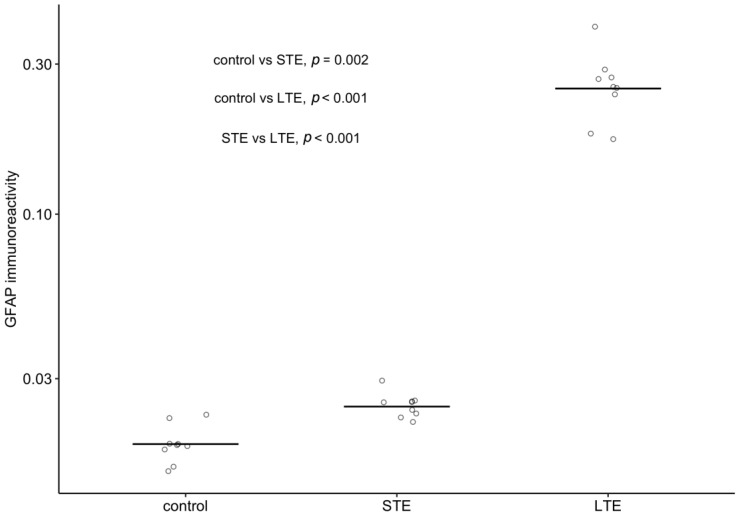
Effect of blue light on the intensity of glial fibrillary acid (GFAP) protein immunoreactivity relative to intensity of DAPI staining in Wistar rat retinas. The rats were exposed to 12 h white light (150 lx) and 12 h darkness for 10 d (control; 9 rats), cycles of 12 h blue light (150 lx) and 12 h darkness for 10 d (long-term exposure, LTE; 9 rats), and constant blue light (150 lx) for 2 d (short-term exposure, STE; 9 rats). Note the logarithmic scale on the vertical axis. Circles show the intensity of GFAP immunoreactivity in one retinal cryosection; horizontal lines show 20%-trimmed means in each group. *p*-values were calculated with Wilcox’s robust method based on 20%-trimmed means [34,35].

**Figure 8 cells-12-01014-f008:**
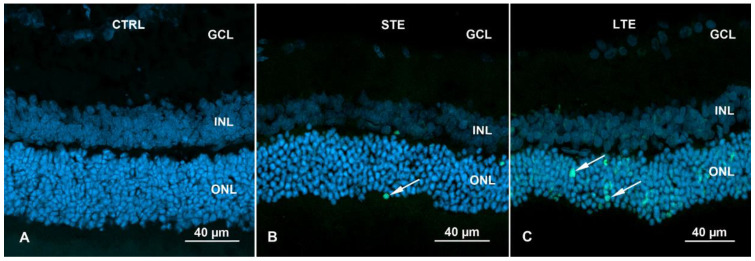
TUNEL assay of Wistar rat retinas. Blue fluorescence shows nuclei stained with DAPI. The rats were exposed to 12 h white light (150 lx) and 12 h darkness for 10 d (control, CTRL; 9 rats), cycles of 12 h blue light (150 lx) and 12 h darkness for 10 d (long-term exposure, LTE; 9 rats), and constant blue light (150 lx) for 2 d (short-term exposure, STE; 9 rats). (**A**) Retina from a rat in the control group. Note: no labelling for apoptosis. (**B**) Retina from a rat in the STE group. Isolated nuclei display labelling for apoptosis (white arrows, green fluorescence) in the ONL. (**C**) Retina from a rat in the LTE group. Note: more labelling for apoptosis is present in the ONL (white arrows, green fluorescence). Samples were visualized and photographed with a confocal microscope (LSM900, Zeiss). Abbreviations: ONL, outer nuclear layer; INL, inner nuclear layer; GCL, ganglion cell layer.

**Figure 9 cells-12-01014-f009:**
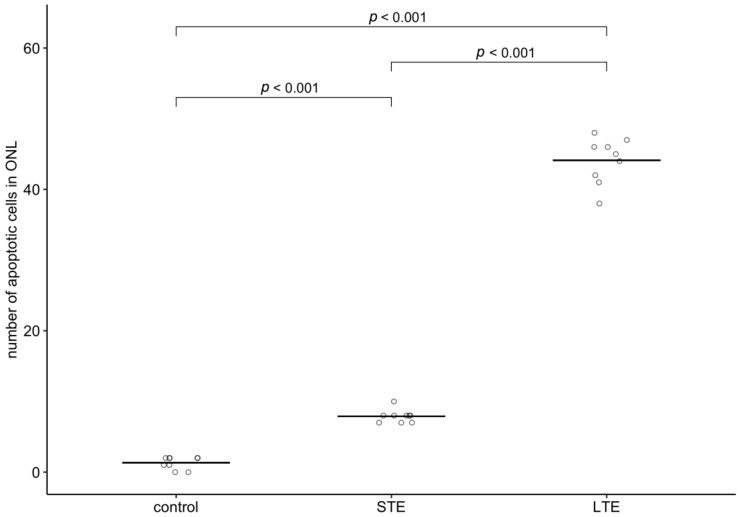
Effect of blue light on the number of apoptotic cells in the outer nuclear layer in Wistar rat retinas. The rats were exposed to 12 h white light (150 lx) and 12 h darkness for 10 d (control, CTRL; 9 rats), cycles of 12 h blue light (150 lx) and 12 h darkness for 10 d (long-term exposure, LTE; 9 rats), and constant blue light (150 lx) for 2 d (short-term exposure, STE; 9 rats). Circles show the intensity of GFAP immunoreactivity in one retinal cryosection; horizontal lines show group means. *p*-values were calculated by quasi-Poisson regression followed by pairwise comparisons (Tukey’s method for *p*-value adjustment).

**Figure 10 cells-12-01014-f010:**
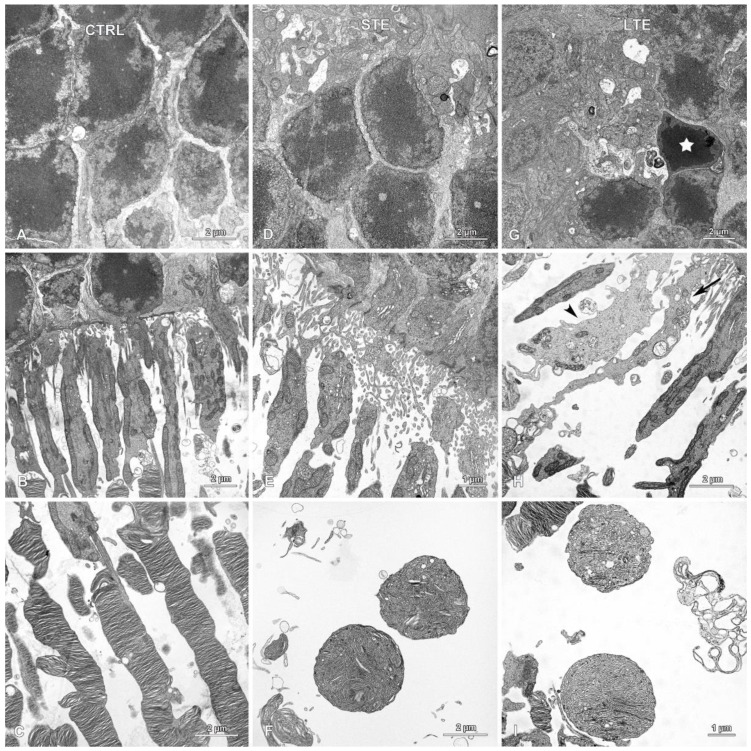
Effect of blue light exposure on the morphology of photoreceptors. Electron micrographs of retinas from Wistar rats. The rats were exposed to 12 h white light (150 lx) and 12 h darkness for 10 d (control, CTRL; 9 rats), cycles of 12 h blue light (150 lx) and 12 h darkness for 10 d (long-term exposure, LTE; 9 rats), and constant blue light (150 lx) for 2 d (short-term exposure, STE; 9 rats). Normal looking outer nuclear layers (ONL) in the (**A**) control and (**D**) STE groups. (**G**) ONL from the LTE group. Note: a single nucleus with pyknotic chromatin (star) present among normal looking nuclei in the ONL. Normal looking photoreceptor outer and inner segments from the (**B**) control and (**E**) STE groups. (**H**) Inner segments from the LTE group display some abnormalities. Note: some inner segments are thinner and longer (arrow), and some are swollen and thicker and have a lighter cytoplasm (arrowhead) than those in the control group. (**C**) Normal looking photoreceptor outer segments in the control group. In the (**F**) STE and (**I**) LTE groups, the apical parts of the photoreceptor outer segments form round or ellipsoidal structures containing tubules and vesicles.

**Figure 11 cells-12-01014-f011:**
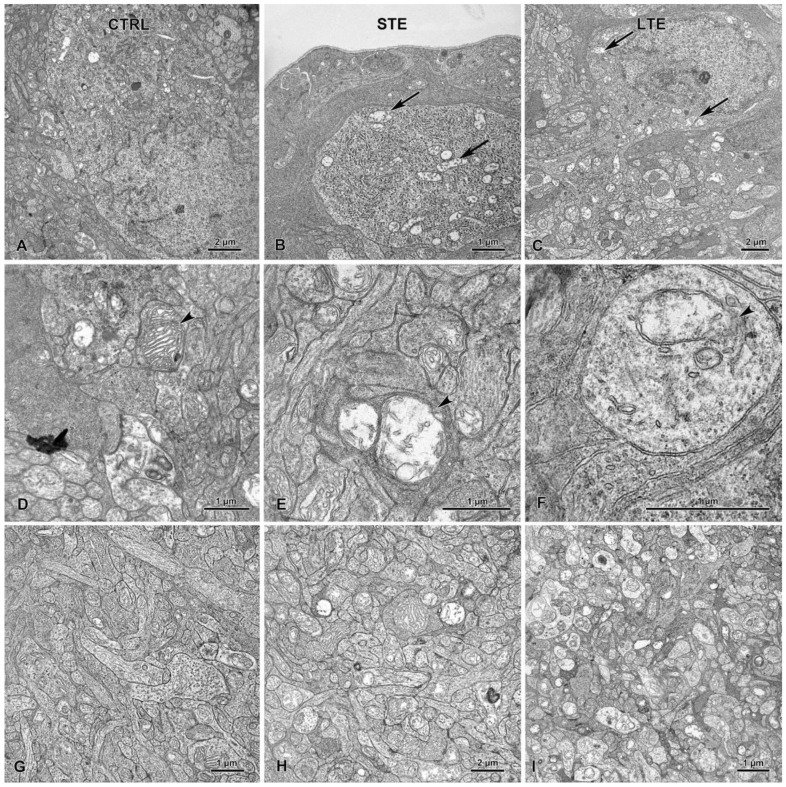
Effect of blue light exposure on the inner retina. The rats were exposed to 12 h white light (150 lx) and 12 h darkness for 10 d (control, CTRL; 9 rats), cycles of 12 h blue light (150 lx) and 12 h darkness for 10 d (long-term exposure, LTE; 9 rats), and constant blue light (150 lx) for 2 d (short-term exposure, STE; 9 rats). The (**A**) control retinal ganglion cells and their mitochondria display normal ultrastructure. In the STE (**B**) and LTE (**C**) groups, all mitochondria are swollen (arrows). Note that control group (**D**) mitochondria display normal ultrastructure in the inner plexiform layer (arrowhead). In the STE (**E**) and LTE (**F**) groups, mitochondria are swollen and have disrupted cristae (arrowheads). The inner plexiform layer appears normal in the control group (**G**). In the STE (**H**) group, some neural processes are swollen, and in the LTE (**I**) group, all processes are swollen.

**Table 1 cells-12-01014-t001:** Assignation of retinas for post-mortem assays.

Animal Number	Whole Mount Immunocytochemistry ^a^						TUNEL ^b^ and ICH ^c^						Transmission Electron Microscopy ^d^					
	Short Term		Long Term		Control		Short Term		Long Term		Control		Short Term		Long Term		Control	
	Right	Left	Right	Left	Right	Left	Right	Left	Right	Left	Right	Left	Right	Left	Right	Left	Right	Left
Rat 1 ♀	+	−	+	−	+	−	−	+	−	+	−	+	−	+	−	+	−	+
Rat 2 ♂	−	+	−	+	−	+	+	−	+	−	+	−	+	−	+	−	+	−
Rat 3 ♀	+	−	+	−	+	−	−	+	−	+	−	+	−	+	−	+	−	+
Rat 4 ♂	−	+	−	+	−	+	+	−	+	−	+	−	+	−	+	−	+	−
Rat 5 ♀	+	−	+	−	+	−	−	+	−	+	−	+	−	+	−	+	−	+
Rat 6 ♂	−	+	−	+	−	+	+	−	+	−	+	−	+	−	+	−	+	−
Rat 7 ♀	+	−	+	−	+	−	−	+	−	+	−	+	−	−	−	−	−	−
Rat 8 ♂	−	+	−	+	−	+	+	−	+	−	+	−	−	−	−	−	−	−
Rat 9 ♀	+	−	+	−	+	−	−	+	−	+	−	+	−	−	−	−	−	−

^a^ For whole-mount immunocytochemistry, the entire retina from one eye was used; ^b^ For terminal deoxynucleotidyl transferase dUTP nick-end labelling (TUNEL) and immunocytochemistry; ^c^ (ICH), the nasal half of the eye was used; ^d^ For transmission electron microscopy, the temporal half of the eye was used.

## Data Availability

Not applicable.

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
