# Peer review of "Low-Intensity Blue Light Exposure Reduces Melanopsin Expression in Intrinsically Photosensitive Retinal Ganglion Cells and Damages Mitochondria in Retinal Ganglion Cells in Wistar Rats"

_cells, 2023, doi:10.3390/cells12071014_

Round 1
Reviewer 1 Report (New Reviewer)
Dear Author,
Thanks for submitting your research manuscript entitled " Low-intensity blue-light exposure reduces melanopsin expression in intrinsically-photosensitive retinal ganglion cells and damages mitochondria in retinal ganglion cells in Wistar rats".
Before giving my final comments as well as the final revision of this manuscript, the author needs to address the following comments scientifically.
Major concerns:
Please find out the following comments:
· The rationale and purpose behind selecting Low-intensity blue-light exposure reduces melanopsin expression in intrinsically-photosensitive retinal ganglion cells is explained very poorly, irrelevant and incomplete manner throughout the manuscript.
· Title and abstract is misleading the reader. Title needs to reframe in simply manner.
· Reviewer surprise to see that in title, author used Wistar rat, and in the end of abstract, they are saying albino rats? Please explain this?
· Theme is very confusing manuscript, one sided using Low-intensity blue-light exposure reduces melanopsin expression and the rationale behind the selection of mitochondria in retinal ganglion cells is another major concern. Authors didn’t explain, and mixing both conditions.
· Required, date and year for animal ethical approval number.
· Lack of update as well old & outdated references with incomplete experimental design is another major concern.
· In Abstract direct statement “This study investigated the effect of low-intensity blue light on the albino rat retina, including intrinsically-photosensitive retinal ganglion cells (ipRGCs). Three groups of nine Wistar rats were used. One group was continuously exposed to blue light (150 lx) for 2 d (STE); one was exposed to 12 h of blue light and 12 h of darkness for 10 d (LTE); one was maintained in 12 h of white light (150 lx) and 12 h of darkness for 10 d (control). Melanopsin (Opn4) was immunolabelled on retinal whole-mounts. To count and measure Opn4-positive ipRGC somas and dendrites (including Sholl profiles), Neuron J was used. Retinal cryosections were immunolabeled for glial fibrillary acid protein (GFAP) and with terminal deoxynucleotidyl transferase dUTP nick-end labelling for apoptosis detection.” Is this abstract? And is confusing. Need to reframe accordingly these types of errors throughout the manuscript.
· This is preliminary investigation. Reason and explanation behind the updation and initiation of this now as research paper?? With these, lack of limitations in current research, especially, on the basis of current evaluation, it is tough to quote “Ultrastructural analysis showed that LTE damaged mitochondria in retinal ganglion cells and in the inner plexiform layer. LTE to low-intensity blue light harms the retinas of albino rats.”. Paper can’t be accepted and encourage to team for further proceeding the methodologies and results. Abstract is very poorly written and very confusing. Irrational and fused with repetitions. The reviewer found irrational and non-scientific justification in the abstract—introduction and discussion part.
- Is this abstract or explanation of results? What authors want to say? The incomplete justification and scientific correlation is another concern.
· The reviewer found irrational and non-scientific justification in the abstract—introduction and discussion part.
· The reviewer feels the author needs to elaborate and justify it with proper citations and strong evidence. The author fails to explain the relevant justification in the introduction as mentioned in the discussion part.
· A major drawback is a lack of supporting clinical as well as pre-clinical evidences regarding cellular and molecular targets involved in the progression of damaging of retinal ganglion cells.
· Throughout the manuscript, the main focus is not clear. Complete mismatch of abstract, introduction, results and discussion.
Title:
· Mismatch of title with relevant introduction and conclusive remarks in the conclusion part.
Abstract:
- The rationale behind this research is not well explained, and several major concerns still constrain the reviewer's enthusiasm for publishing this manuscript.
Introduction:
- The basic literature is not well written and does not even include any literature on alternative approaches with updated references regarding involvement of current drug treatment/techniques used in pathogenesis and development of retinal ganglion cells associated neuropathological illnesses.
- Authors fail to justify the correlation, and almost irrational and common information is present in the introduction part.
Material and methods:
- Major drawback is the lack of supporting references and incomplete experimental and paradigms.
- A separate detail paragraph required here, to explain the experimental design for rodents with detail explanation in flow chart also.
- All biochemical parameters are poorly explained without any sense, without any references.
- Provide all biochemicals kits numbers along with their city, country in all individual parameters in all expressions, blots, etc.
- In order to support the assessment of all mentioned parameters in his study, the author should provide all the source documents and data he/she has followed for all assays and estimates.
- How was the sample size determined? Ideally, a priori sample size calculation should be performed to determine the appropriate sample size.
- Normality and variance homogeneity should be assessed across all groups of the same outcome variable and not individual experimental groups. If the data were not normally distributed or variance homogeneity was not met, nonparametric tests need to be performed. Parametric data should be reported as mean +/- SD, while nonparametric data should be given/displayed as median and interquartile range. Longitudinal data should be analyzed using repeated measures tests.
Results:
- All results are very-very poorly explained. Revised explanation for all as above mentioned instructions.
- Figure 10, without scale and bars. Need to add accordingly.
- Results need more clarification and significant justification. Differentiating between the outcome and the discussion sections is quite difficult.
- High note: Must provide all results description and Use proper statistical reporting: i.e. for the results of each statistical test, the authors should report the statistical test that was applied, the test statistic (e.g. t, U, F, r), degrees of freedom as subscripts to the test statistic, and the exact probability value, including those for normality and variance homogeneity tests. Statistics should be reported in APA format, i.e.: t(df) = value, p = value; F(df1,df2) = value, p = value; r(df) = value, p = value; [chi]2 (df, N = value) = value, p = value; Z = value, p = value. Include statements on the tests for normality and variance heterogeneity and respective results. If the data were not normally distributed or variance heterogeneity was not met, nonparametric tests need to be applied.
Discussion:
- To address the outcome of in-vivo measures/results separately avoiding the hyperglycemia and peripheral insulin resistance and maintaining physiological condition and how they correlate with the existing literature, it would be better if the author restructured to take a more critical approach for effective in Low-intensity blue-light exposure reduces melanopsin expression conditions.
- In the discussion and the conclusion, the aims, rationale, and future perspectives are not evident clearly in relation with in-vitro and in-vivo experimentation.
- The discussion is usually unorganized at the beginning to address all the observations and evaluate them at the end. It makes the results easier to contextualize and simpler to comprehend.
- Furthermore, a minimal critical analysis should be provided, along with current study limitations as well the future perspective as separate paragraph.
Conclusion:
- Need to revise the conclusion in a scientific manner. Not accepted in its current form.
- This reviewer considers that this paper cannot be published in the present form. A detailed revision shortening, ordering and following the commented ideas could improve this interesting paper in a significant manner.
- Several typewriting mistakes are present and needing correction. This reviewer remains at entire disposal for the next version.
Author Response
The Authors would like to thank the associate editor and reviewer for review of our manuscript and for their comments and suggestions for improving the quality of the manuscript. The following responses have been prepared to address the reviewer’s comments in a point-by-point fashion.
Reviewer #1:
Major concerns:
Please find out the following comments:
Comment:
- The rationale and purpose behind selecting Low-intensity blue-light exposure reduces melanopsin expression in intrinsically-photosensitive retinal ganglion cells is explained very poorly, irrelevant and incomplete manner throughout the manuscript.
Authors’ response:
Thank you for this suggestion. The rationale for performing this study has been expanded by adding information in lines 45-57.
Comment:
- Title and abstract is misleading the reader. Title needs to reframe in simply manner.
Authors’ response:
We are puzzled by this comment. The title summarizes the main findings of this study, i.e., (1) Low-intensity blue-light exposure reduces melanopsin expression in intrinsically-photosensitive retinal ganglion cells and (2) damages mitochondria in retinal ganglion cells in Wistar rats. We are wondering why the reviewer thinks that these two simple statements are misleading, and we request specific suggestions for improvement.
Comment:
- Reviewer surprise to see that in title, author used Wistar rat, and in the end of abstract, they are saying albino rats? Please explain this?
Authors’ response:
Thank you for drawing our attention to this. To clarify, we used albino Wistar rats. This has now been specified in the Abstract (lines 14, 15, 28).
Comment:
- Theme is very confusing manuscript, one sided using Low-intensity blue-light exposure reduces melanopsin expression and the rationale behind the selection of mitochondria in retinal ganglion cells is another major concern. Authors didn’t explain, and mixing both conditions.
Authors’ response:
Thank you for suggesting that we better explain our rationale for examining retinal ganglion cells. To implement this suggestion, we have added the following to explain why we examined mitochondria in retinal ganglion cells (lines 69-75 ): “There are number of studies demonstrating that short-wavelength light (400-470nm) is absorbed by enzyme complexes of the electron transport system in mitochondria and that it negatively affects mitochondria function in photoreceptors and RGCs [27-32]. Moreover, exposure to blue light inhibits mitochondrial enzymes related to the oxidative cycle, which can lead to synthesis of reactive oxygen species that are toxic to the retina [31-32]. Thus, it seems important to investigate whether exposure to low-intensity blue light will affect the mitochondria present in RGCs, which contain a large number of these organelles [33-34].”
“Additionally, although electron microscopy does not differentiate between ipRGCs and other RGCs, we used this technique to investigate the ultrastructure of RGCs for better insight into the effect of this exposure on the morphology of the RGC layer.” (lines 80-83)
As for the other parts of the reviewer’s comment, we are unsure what he/she meant. More specifically, we do not understand how the “theme is… [a] manuscript”, how “one-sided” uses “Low-intensity blue-light exposure reduces melanopsin expression”, or what conditions were mixed. We would welcome clarification of these points by the reviewer and specific suggestions for improvement.
Comment:
- Required, date and year for animal ethical approval number.
Authors’ response:
This information has been added to the methods section (line 106).
Comment:
- Lack of update as well old & outdated references with incomplete experimental design is another major concern.
Authors’ response:
The reference list has been updated. Although four of our references are from before 2000, they are important for providing context for the present results: Kuwabara 1968, Ham 1976, papers from Burns et al. 1990, and De Raad et al. 1996; therefore, we would like to keep these references in the reference list. All of the other cited papers are related to the topic of this paper and have been published after 2000. If the reviewer would like to suggest any specific references, the authors would be grateful.
Comment:
- In Abstract direct statement “This study investigated the effect of low-intensity blue light on the albino rat retina, including intrinsically-photosensitive retinal ganglion cells (ipRGCs). Three groups of nine Wistar rats were used. One group was continuously exposed to blue light (150 lx) for 2 d (STE); one was exposed to 12 h of blue light and 12 h of darkness for 10 d (LTE); one was maintained in 12 h of white light (150 lx) and 12 h of darkness for 10 d (control). Melanopsin (Opn4) was immunolabelled on retinal whole-mounts. To count and measure Opn4-positive ipRGC somas and dendrites (including Sholl profiles), Neuron J was used. Retinal cryosections were immunolabeled for glial fibrillary acid protein (GFAP) and with terminal deoxynucleotidyl transferase dUTP nick-end labelling for apoptosis detection.” Is this abstract? And is confusing. Need to reframe accordingly these types of errors throughout the manuscript.
Authors’ response:
Unfortunately, we found it difficult to respond to this comment due to the lack of specific suggestions by the reviewer. Moreover, we are confused because the words quoted by the reviewer seem to fulfill the Instructions for Authors: “…highlight the purpose of the study; 2) Methods: Describe briefly the main methods or treatments applied. Include any relevant preregistration numbers, and species and strains of any animals used.” (https://www.mdpi.com/journal/cells/instructions)
The Abstract was written with clarity in mind and to conform to the word limit of approximately 200 words imposed by the journal. This was done in close cooperation with a native-speaker with extensive experience in the field and using the guidelines in well-respected books on scientific writing in English (Zeiger 2000, p. 269–90 and Hofmann 2014, p. 310–320). We used short, active, direct statements with parallel structures because they are recommended for scientific English (Zeiger 2000, p. 22–26, 33–34, 40–45; Hofmann 2014, p. 48–56, 59–63).
If the reviewer could expand on his/her general criticisms by making specific suggestions for improving the Abstract while respecting the word limit, we would be grateful.
References:
Hofmann, A: Scientific Writing and Communication. 2014, Oxford University Press.
Zeiger M. Essentials of Writing Biomedical Research Papers. Second Edition. 2000, McGraw Hill / Medical.
Comment:
- This is preliminary investigation. Reason and explanation behind the updation and initiation of this now as research paper?? With these, lack of limitations in current research, especially, on the basis of current evaluation, it is tough to quote “Ultrastructural analysis showed that LTE damaged mitochondria in retinal ganglion cells and in the inner plexiform layer. LTE to low-intensity blue light harms the retinas of albino rats.”. Paper can’t be accepted and encourage to team for further proceeding the methodologies and results. Abstract is very poorly written and very confusing. Irrational and fused with repetitions. The reviewer found irrational and non-scientific justification in the abstract—introduction and discussion part.
Authors’ Response:
We were confused by this comment and would welcome clarification on the part of the reviewer.
- What was meant by “updation and initiation of this now as research paper”?
- We are perplexed because we did not quote anything in this article. All writing is our own original statements, including the words “Ultrastructural analysis showed that LTE damaged mitochondria in retinal ganglion cells and in the inner plexiform layer. LTE to low-intensity blue light harms the retinas of albino rats”. Also, what is the relationship between “limitations in current research” and our statements based on the evidence in our own study? Please clarify what was meant here.
- Please specify exactly what was irrational. We were unable to find what the reviewer was referring to.
- Repetition of key words is considered good style in scientific English because it improves clarity (Zeiger 2000, p. 58–63; Hofmann 2014, p. 106–108 [see references given above]). Thus, we would like to keep such repetitions in our manuscript.
As mentioned above, we would be grateful for any specific examples of what the reviewer’s statements refer to.
Comment:
- Is this abstract or explanation of results? What authors want to say? The incomplete justification and scientific correlation is another concern.
Authors’ Response:
We are confused by this comment because the Abstract is clearly labeled “Abstract”, and the Instructions for Authors indicate that it should provide the key results of the study (https://www.mdpi.com/journal/cells/instructions). The word limit of 200 words made it impossible to provide more background information in the Abstract without sacrificing key results and experimental detail. If the reviewer has a specific suggestion for fitting in more background information without sacrificing key results and experimental detail, we would welcome it.
Please explain what was meant by “scientific correlation” here. We did not perform any correlation analysis, and none of our research questions concerned hypotheses that could be investigated with correlation analysis.
Comment:
- The reviewer found irrational and non-scientific justification in the abstract—introduction and discussion part.
- The reviewer feels the author needs to elaborate and justify it with proper citations and strong evidence. The author fails to explain the relevant justification in the introduction as mentioned in the discussion part.
Authors’ Response
As mentioned above, the rationale for performing this study has been expanded (lines 45-57).
Comment:
- A major drawback is a lack of supporting clinical as well as pre-clinical evidences regarding cellular and molecular targets involved in the progression of damaging of retinal ganglion cells.
Authors’ response:
Unfortunately, we were confused by the reference to clinical evidence. This sort of evidence concerns the effect of treatments, not specific molecular targets, and we are not conducting a clinical trial. Please explain what was meant here.
In an attempt to provide what the reviewer requested, we have added information about molecular targets that may be involved in the progression of ganglion cell damage in the discussion (lines 553-558). We would be happy to consider any other references that the reviewer can suggest to further improve this aspect of our manuscript.
Comment:
- Throughout the manuscript, the main focus is not clear. Complete mismatch of abstract, introduction, results and discussion.
Title:
- Mismatch of title with relevant introduction and conclusive remarks in the conclusion part.
Abstract:
- The rationale behind this research is not well explained, and several major concerns still constrain the reviewer's enthusiasm for publishing this manuscript.
Authors’ Response
We are eager to improve our manuscript as much as possible, but the general nature of some of these comments meant that we were unsure what the reviewer was suggesting. Moreover, this comment seems to conflict with the other reviewer’s assessment that “The paper is well written, the results are generally clear, and the discussion does not overinterpret the results.” Specific examples of mismatches between the Title, Abstract, Introduction, Results, Discussion, and Conclusion would be welcomed.
As mentioned above, the rationale for performing this study has been expanded.
Comment:
Introduction:
- The basic literature is not well written and does not even include any literature on alternative approaches with updated references regarding involvement of current drug treatment/techniques used in pathogenesis and development of retinal ganglion cells associated neuropathological illnesses.
Authors’ response:
This is an interesting comment, but unfortunately, we are unsure how it applies to our study. Our study concerns the effect of blue-light exposure on the retina, with a focus on ipRGCs. We were not investigating RGCs in the context of neuropathological illnesses. Further explanation would be appreciated.
Comment:
- Authors fail to justify the correlation, and almost irrational and common information is present in the introduction part.
Authors’ response:
We are confused by this comment because no correlation analyses were performed in this study; thus, further explanation would be much appreciated. We would welcome any details about which pieces of information are irrational, so that we can correct these parts of the Introduction. Finally, we cite recent literature and literature that provides the necessary background for readers of Cells to understand our study, so we do not understand what was meant by “common information.” We would be grateful for any clarifications.
Comment:
Material and methods:
- Major drawback is the lack of supporting references and incomplete experimental and paradigms
- A separate detail paragraph required here, to explain the experimental design for rodents with detail explanation in flow chart also.
Authors’ response:
Thank you for the suggestion of a flow chart, it has been added to supplement the information in section 2.1 (Animals and experimental design). We would like to point out that information about experimental design and the rodents is already given in two paragraphs at the beginning of the Methods section (lines 85-107).
Comment:
- All biochemical parameters are poorly explained without any sense, without any references.
Authors’ response:
We respectfully ask for an explanation of the term “biochemical parameters” in this context because we did not examine biochemistry. Instead, as stated at the end of the Introduction (lines 77-83), “Our objective was to investigate the effects of short-term (2 d) and long-term (10 d) exposure to low-intensity blue light (150 lx) on the retinas of Wistar rats. Specifically, we investigated the effect of exposure to this kind of light on the expression of Opn4 in ipRGCs and on retinal morphology in whole mount retinas. Additionally, although electron microscopy does not differentiate between ipRGCs and other RGCs, we used this technique to investigate the ultrastructure of RGCs for better insight into the effect of this exposure on the morphology of the RGC layer.” The importance of Opn4 is explained in two paragraphs in the Introduction (lines 47-57 and 58-68).
Comment:
- Provide all biochemicals kits numbers along with their city, country in all individual parameters in all expressions, blots, etc.
Authors’ response:
Thank you for this suggestion. This information has been provided for all important chemicals used in this study.
Comment:
- In order to support the assessment of all mentioned parameters in his study, the author should provide all the source documents and data he/she has followed for all assays and estimates.
Authors’ response:
We respectfully request clarification of what was meant by this comment. We used our assays to produce our data, and the data was used to produce our estimates, and we believe that all of the information about how this was done is already included in the article. If this is not the case, please indicate exactly what should be improved.
Comment:
- How was the sample size determined? Ideally, a priori sample size calculation should be performed to determine the appropriate sample size.
Authors’ response:
We agree that a priori sample size calculation is the ideal for confirmatory studies. However, this is not a confirmatory study, and sample size calculation requires estimates of the group means, standard deviations, etc., which were not available until we had performed our study. Thus, we did not perform these calculations.
Comment:
Normality and variance homogeneity should be assessed across all groups of the same outcome variable and not individual experimental groups. If the data were not normally distributed or variance homogeneity was not met, nonparametric tests need to be performed. Parametric data should be reported as mean +/- SD, while nonparametric data should be given/displayed as median and interquartile range. Longitudinal data should be analyzed using repeated measures tests.
Authors’ response:
For several reasons, we are unsure how this comment applies to our study, and we ask for further explanation from the reviewer. First, we indicated that the normality and homogeneity of variance of each outcome variable were assessed across all experimental groups. As stated in section 2.7, “Before analysing the lengths and counts of Opn4-positive neurons and the intensity of GFAP immunoreactivity, the data were checked to verify whether the assumptions of normality and homogeneity of variance were met. This was done using normal quantile-quantile plots and scale-location plots, respectively (lines 160-163).” With regard to the Sholl data, “Diagnostic plots showed that the data was approximately normally distributed with homogenous variance, and influence diagnostics indicated that no outliers had an undue influence on the results (lines 175-177).”
Second, when it appeared that the data were not normally distributed or variance was not homogenous, we used modern robust methods that are significant improvements over older, rank-based “nonparametric” methods. (See Wilcox and Rousselet (2018) for a discussion of problems with older approaches and the advantages of modern improvements, including decreased false-positive and false-negative rates.)
Third, because these modern methods use 20%-trimmed means in order to preserve information that is lost when using medians and rank-based approaches, we report the 20%-trimmed means when appropriate, not means or medians. Moreover, we show every data point in our figures (except for Figure 4, which shows the range, interquartile range, mean, and median), which provides even more information than a summary statistic like the standard deviation.
Finally, no longitudinal data was analyzed in this study. Regarding the use of repeated measures tests for analysis of Sholl data, we prefer to use mixed models as opposed to repeated-measures tests because these models have many advantages over the older repeated-measures methods and are recommended when analyzing Sholl data (Wilson et al. 2017).
Wilcox RR, Rousselet GA. A guide to robust statistical methods in neuroscience. Current protocols in neuroscience. 2018 Jan;82(1):8-42.
Wilson MD, Sethi S, Lein PJ, Keil KP. Valid statistical approaches for analyzing Sholl data: Mixed effects versus simple linear models. Journal of neuroscience methods. 2017 Mar 1;279:33-43.
Comment:
Results:
- All results are very-very poorly explained. Revised explanation for all as above mentioned instructions
- Figure 10, without scale and bars. Need to add accordingly.
- Results need more clarification and significant justification. Differentiating between the outcome and the discussion sections is quite difficult.
- High note: Must provide all results description and Use proper statistical reporting: i.e. for the results of each statistical test, the authors should report the statistical test that was applied, the test statistic (e.g. t, U, F, r), degrees of freedom as subscripts to the test statistic, and the exact probability value, including those for normality and variance homogeneity tests. Statistics should be reported in APA format, i.e.: t(df) = value, p = value; F(df1,df2) = value, p = value; r(df) = value, p = value; [chi]2 (df, N = value) = value, p = value; Z = value, p = value. Include statements on the tests for normality and variance heterogeneity and respective results. If the data were not normally distributed or variance heterogeneity was not met, nonparametric tests need to be applied.
Authors’ response:
Regarding the statistical methods, we did provide the name of each statistical test in section 2.1 (i.e., “quasi-Poisson regression” and “a mixed model with a random intercept for each rat followed by Tukey’s method (with Satterthwaite degrees of freedom)”). We have added an explanation that Wilcox’s robust method is a generalization to trimmed means of Dunnett’s method (line 163-167), plus one more reference (Mair and Wilcox 2020), so that readers do not need to access Wilcox’s textbook to understand this method.
As for the suggestion to use formal tests of deviations from normality and homogeneity of variance, this would be an excellent suggestion for users of statistical packages that do not yet provide improved methods for dealing with violations of these assumptions. However, these tests have many problems, as explained in many statistical textbooks (e.g., Kirkwood and Sterne 2003, p. 112; Wilcox 2017, p. 269–70; Motulsky 2018, p. 224–230) and discussed at length in an article by Wilcox and Rousselet (2018). There are now better options available in R and Python (Wilcox and Rousselet (2018), Mair and Wilcox (2020), Campopiano and Wilcox (2020)). As explained in section 2.1 of the manuscript, we used these improved modern methods, and we would like to keep them because they substantially reduce both false-negative and false-positive results. To further help our readers understand and access these improved methods, we have added exact details of the statistical software we used.
Unfortunately, we were unable to implement the other suggestions because we need further clarification from the reviewer. First, please explain where the “above mentioned instructions” for the Results section are.
Second, Figure 10 already has scale bars, so we do not understand what we should do.
Third, please explain how we should clarify and justify the results and differentiate between the outcome and discussion sections. Note that, in accordance with the guidelines in Zeiger (2000, p. 183–187) and Hofmann (2014, p. 286–289), we do provide a short summary of the main findings at the beginning of the Discussion section. We would like to keep this summary there because, according to those experts on scientific communication, it is helpful for many readers.
Fourth, we believe that all results have been described. Thus, we ask the reviewer to please explain what he/she thinks is missing.
Finally, we are unsure why the reviewer is asking us to follow the American Psychological Association (APA) guidelines for reporting statistics when we follow the SAMPL guidelines for biomedical publications (Lang and Altman, 2013), which seem more appropriate for a biomedical publication in Cells. Moreover, the information about APA format that the reviewer provided does not apply to any of the statistical procedures that we used.
References:
Campopiano A, Wilcox RR. Hypothesize: robust statistics for Python. Journal of Open Source Software. 2020 Jun 19;5(50):2241.
Hofmann, A: Scientific Writing and Communication. 2014, Oxford University Press.
Kirkwood BR, Sterne JAC. Essential medical statistics, 2nd ed. Blackwell Science; 2003.
Mair P, Wilcox R. Robust statistical methods in R using the WRS2 package. Behavior research methods. 2020 Apr;52:464-88.
Motulsky H. Intuitive biostatistics: a nonmathematical guide to statistical thinking, 4th ed. Oxford University Press, USA; 2018.
Thomas A. Lang, Douglas G. Altman, Basic statistical reporting for articles published in Biomedical Journals: The “Statistical Analyses and Methods in the Published Literature” or the SAMPL Guidelines, International Journal of Nursing Studies, Volume 52, Issue 1, 2015, Pages 5-9, ISSN 0020-7489, https://doi.org/10.1016/j.ijnurstu.2014.09.006.
Wilcox RR. Understanding and applying basic statistical methods using R. John Wiley & Sons; 2017.
Wilcox RR, Rousselet GA. A guide to robust statistical methods in neuroscience. Current protocols in neuroscience. 2018 Jan;82(1):8-42.
Zeiger M. Essentials of Writing Biomedical Research Papers. Second Edition. 2000, McGraw Hill / Medical.
Comment:
Discussion:
- To address the outcome of in-vivo measures/results separately avoiding the hyperglycemia and peripheral insulin resistance and maintaining physiological condition and how they correlate with the existing literature, it would be better if the author restructured to take a more critical approach for effective in Low-intensity blue-light exposure reduces melanopsin expression conditions.
Authors’ response:
We are unsure how hyperglycemia and peripheral insulin resistance relate to our manuscript because these topics were completely beyond the scope of our study. If the reviewer could indicate exactly how these topics relate to our results, it would be much appreciated.
Comment:
- In the discussion and the conclusion, the aims, rationale, and future perspectives are not evident clearly in relation with in-vitro and in-vivo experimentation.
- The discussion is usually unorganized at the beginning to address all the observations and evaluate them at the end. It makes the results easier to contextualize and simpler to comprehend.
Authors’ response:
Unfortunately, we are uncertain what the first comment refers to, as there was no in-vitro experimentation in this study, and we are unsure what the reviewer meant by “are not evident clearly in relation with”. As for the second comment, we did present all of the important findings at the beginning of the discussion, then evaluate them in later parts of the discussion; thus, it seems that we have already done what the reviewer is asking for. We would be grateful for any clarification of these points.
Comment:
- Furthermore, a minimal critical analysis should be provided, along with current study limitations as well the future perspective as separate paragraph.
Authors’ response:
Thank you for this suggestion. To address these points, a paragraph has been added to the end of the Discussion, and two sentences have been added to the Conclusions. (lines 576-585, 596-601)
Comment:
Conclusion:
- Need to revise the conclusion in a scientific manner. Not accepted in its current form.
Authors’ response:
Unfortunately, the general nature of this comment meant that we were unclear on what the reviewer was requesting. We can state that the Conclusion was written in accordance with the guidance in Zeiger (2000, p. 196–199) and Hofmann (2014, p. 294–296) (complete references given above) and add that the other reviewer stated that “The paper is well written, the results are generally clear, and the discussion does not overinterpret the results.” However, in an attempt to satisfy this reviewer, two sentences have been added to the Conclusions (lines 596-601).
Comment:
- This reviewer considers that this paper cannot be published in the present form. A detailed revision shortening, ordering and following the commented ideas could improve this interesting paper in a significant manner.
Authors’ response:
We are uncertain as to what the reviewer wants us to shorten and reorganize, as the reviewer’s comments have only indicated that we should add more information to the paper or change what was written without shortening it, and none of the comments have touched on reorganization. We would like to reiterate that the paper was written in accordance with the guidance in Zeiger (2000) and Hofmann (2014) (complete references given above) and that the second reviewer stated that “The paper is well written, the results are generally clear, and the discussion does not overinterpret the results.” We have done our best to implement all of the first reviewer’s comments, and where that proved impossible because of their general nature, we have respectfully requested more specific details. We remain eager to improve our manuscript as much as possible, so any specific suggestions would be greatly appreciated.
Comment:
- Several typewriting mistakes are present and needing correction. This reviewer remains at entire disposal for the next version.
Authors’ response:
The manuscript has been carefully checked by a native speaker of English with extensive experience in the field. In the event that any typographical errors remain, we would be grateful for an indication of what they are and where they can be found.

Reviewer 2 Report (New Reviewer)
The authors present data indicating that prolonged blue light exposure damages intrinsically-photosensitive retinal ganglion cells and the mitochondria in rats. The paper is well written, the results are generally clear, and the discussion does not overinterpret the results. There are just a few minor items that the authors should attend to.
1) Fig 1: Although the statistical analysis is well described and seems appropriate, the authors should acknowledge that the lack of P < 0.05 is probably due to an inadequate # of animals.
2) Along the same lines, authors should indicate if animals were assigned randomly to groups and the # of and reasons for any dropouts. Did you originally estimate n = 9 per group and if so, how did you arrive at that estimate?
3) Figure 6 appears to show higher GFAP expression in controls (according to the representative picture) vs the STE group. How do the authors explain this finding?
Author Response
The Authors would like to thank the associate editor and reviewer for review of our manuscript and for their comments and suggestions for improving the quality of the manuscript. The following responses have been prepared to address the reviewer’s comments in a point-by-point fashion.
Reviewer # 2
The authors present data indicating that prolonged blue light exposure damages intrinsically-photosensitive retinal ganglion cells and the mitochondria in rats. The paper is well written, the results are generally clear, and the discussion does not overinterpret the results. There are just a few minor items that the authors should attend to.
Comment:
1) Fig 1: Although the statistical analysis is well described and seems appropriate, the authors should acknowledge that the lack of P < 0.05 is probably due to an inadequate # of animals.
Authors’ response:
Thank you for pointing this out. This has been added to a paragraph on study limitations at the end of the Discussion. (lines 576-585)
Comment:
2) Along the same lines, authors should indicate if animals were assigned randomly to groups and the # of and reasons for any dropouts. Did you originally estimate n = 9 per group and if so, how did you arrive at that estimate?
Authors’ response:
Information about the random assignation of rats and the fact that there were no dropouts has been added to the Methods (lines 93–94). We did not perform a priori sample size calculations because we did not have good estimates of central tendencies (e.g., group means) and variation (e.g., standard deviations) under these conditions.
Comment:
3) Figure 6 appears to show higher GFAP expression in controls (according to the representative picture) vs the STE group. How do the authors explain this finding?
Authors’ response:
We are grateful for this comment. To explain, the following has been added to the Results section: “When assessing the strength of immunoreactivity, it is important to keep in mind that it should be assessed relative to a baseline intensity of staining because simple visual inspections can be misleading. Therefore, readers are directed to Figure 6 for the distribution of GFAP immunoreactivity and to Figure 7 for the intensity of its immunoreactivity relative to that of DAPI.” (Lines 291–295)

Round 2
Reviewer 1 Report (New Reviewer)
Dear Author,
After careful review, revised manuscript can proceed further for publication.
This manuscript is a resubmission of an earlier submission. The following is a list of the peer review reports and author responses from that submission.
Round 1
Reviewer 1 Report
The work of Ziólkowska et al. studies the effect of exposure to different blue light intensities on the albino rat retina and, in particular, but not restricted to, on the population of intrinsically photosensitive cells. The paper is generally well written and the experiment well designed. However, there are some aspects that in the opinion of this reviewer should be clarified.
In the methods section, it would be important to clarify the number of animals used. On the one hand, it is important for the authors to specify the number of animals and samples analyzed in each experiment. For example, indicate the number of eyes and sections per eye observed/quantified in each experiment. If this is consistent, it would be sufficient to describe it in the methodology. In addition, the authors indicate that 24 animals were used (12 per sex) but then describe 3 groups of 9. Please clarify further the number of animals and groups used as well as the number of animals per group.
How many eyes were used for retinal whole mounts and how many for cross-sections?
The authors should also elaborate on the quantification of ipRGCs: was it done manually, on how many retinas per group, how many investigators performed the measurements, on whole retinas? It would be desirable that they compare their results with those of previous work quantifying ipRGCs (for example: 10.1167/iovs.17-23258) and indicate whether or not they correlate and the causes of possible differences. The same for the length of Opn4-positive ipRGCs processes. Again, indicate the number of animals/eyes/retinas used for this quantification.
The explanation of the techniques used for the photographing and analysis of the samples is totally missing.
Another point to take into account, which is not clearly stated by the authors, is the age of the animals. Were they all the same age at the beginning of the experiment? Which one?
In addition, the experimental design does not clearly specify the time of animal sacrifice - were all animals sacrificed immediately at the end of the light exposure? Did the authors wait a few days? Also, at what time of day were the animals sacrificed? It is known that the latter can affect melanopsin expression, were they all sacrificed at the same time of day?
How does the photoreceptor impairment compare with other work that can be found in the literature studying the effects of blue light in this population?
Regarding the TUNEL+ cells, since the authors have the quantifications, it would help the understanding of the results if they could incorporate a graph showing the apoptotic cells by zones and if they could perform the statistical analysis between the different groups. For the latter, it would be key to clearly indicate, as requested for the methods section, the number of animals, eyes and sections per eye that have been dedicated to the study of apoptosis (as for the rest of the analyses).
The data shown on GFAP overexpression is very interesting but is purely qualitative. It would help if the authors could provide quantitative data, either through WB or through the measurement of GFAP signal intensity in the sections through some free software such as ImageJ, which is something already described in the literature and which would add more value to the results shown in this work.
In the results section, it would facilitate the reader's understanding if the authors could begin each sub-section by describing the general findings of the control group and the other groups and then indicate how they compare.
Finally, and as indicated in some of the comments, this reviewer considers that the authors have left uncited much of the recent and relevant literature on this topic that could enrich, mainly, the discussion section. As it would provide comparisons of how the retina is affected by exposure to light, both in terms of photoreceptor involvement and glial reaction and ipRGC involvement, this reviewer recommends that the authors review the most recent literature and include it in order to enrich the discussion of the paper.
Reviewer 2 Report
In this study the authors have determined the effect of low-intensity blue light on retinal morphology with a focus on melanopsin (Opn4) expressing retinal ganglion cells (RGCs). Overall the study is qualitative and lacks quantification of most results. The study also does not provide the granularity to rule out whether the deleterious effects of low intensity blue light on Opn4 RGCs is a bystander effect when the whole retina is undergoing degeneration or is there any specificity. The study needs to be bolstered with more experiments and quantification before it can be published in any journal. Below are my specific comments:
1. Melanopsin RGCs come in many types which can differ in their overall morphology of soma, neurites as well as their lamination. The authors can focus on the Opn4 RGCs whose neurites laminate more in the deeper or superficial layers of the inner synaptic layer. This will improve the comparison of neurite length or any such morphological feature
2. The authors must perform sholl analysis to look at the branching and complexity of the dendritic arbor which will be a better parameter for quantify the deleterious effects of blue light.
3. The authors need to include comparison with 1-2 other non-ipRGC types to show whether the morphological changes are occurring in general for all RGCs. Also does the total density of RGCs change? These are important controls.
4. Fig 1. cell counts should be converted into density (number per unit area)
5. Fig 4-6 needs some quantitative comparison of fluorescence or cell counts across conditions. It is very qualitative. Are there selective effects on rods vs cones?
6. Given the impact on the photoreceptor morphology after STE and LTE (Fig 4), visual function should be severely affected. Can the authors do ERG or some test for visual function to determine this? Moreover, if the ipRGCs are being affected is the pupillary light reflex or the circadian rhythm of these rats altered?
7. The importance of studying the effect of low intensity blue intensity on the retina needs to be better motivated.